# Targeted delivery of CRISPR-Cas9 ribonucleoprotein into arthropod ovaries for heritable germline gene editing

Duverney Chaverra-Rodriguez[1], Vanessa M. Macias[1], Grant L. Hughes[1,2], Sujit Pujhari[1], Yasutsugu Suzuki[1,3], David R. Peterson[1], Donghun Kim[1], Sage McKeand[1] & Jason L. Rasgon[1]

Cas9-mediated gene editing is a powerful tool for addressing research questions in arthropods. Current approaches rely upon delivering Cas9 ribonucleoprotein (RNP) complex by embryonic microinjection, which is challenging, is limited to a small number of species, and is inefficient even in optimized taxa. Here we develop a technology termed Receptor-Mediated Ovary Transduction of Cargo (ReMOT Control) to deliver Cas9 RNP to the arthropod germline by injection into adult female mosquitoes. We identify a peptide (P2C) that mediates transduction of Cas9 RNP from the female hemolymph to the developing mosquito oocytes, resulting in heritable gene editing of the offspring with efficiency as high as 0.3 mutants per injected mosquito. We demonstrate that P2C functions in six mosquito species. Identification of taxa-specific ovary-specific ligand–receptor pairs may further extend the use of ReMOT Control for gene editing in novel species.

[1] Department of Entomology, The Huck Institutes of the Life Sciences, and the Center for infectious Disease Dynamics, The Pennsylvania State University, University Park, PA 16802, USA. [2] Present address: Department of Pathology, Institute for Human Infections and Immunity, Center for Tropical Diseases, Center for Biodefense and Emerging Infectious Disease, University of Texas Medical Branch, Galveston, TX 77555, USA. [3] Present address: Institut Pasteur, Viruses and RNA Interference Unit, CNRS Unité Mixte de Recherche, 3569 Paris, France. These authors contributed equally: Duverney Chaverra-Rodriguez, Vanessa M. Macias, Grant L. Hughes Correspondence and requests for materials should be addressed to J.L.R. (email: jlr54@psu.edu)

Since the first application of the Clustered Regularly Interspaced Short Palindromic Repeats and its associated endonuclease 9 (CRISPR-Cas9 system) for site-specific genome editing, the technology has been used in a variety of arthropod species[1,2]. Successful application of CRISPR-Cas9 to edit the germline of arthropods relies on injection of gene-editing materials into pre-blastoderm embryos (embryonic microinjection)[3–6]. This dependence is a significant barrier to the successful general application of transformation technologies, particularly for non-specialist laboratories, as it requires expensive equipment and training to implement[4]. In addition, many non-model species are recalcitrant to the technique because their eggs are damaged during the injection procedure, because they cannot be induced to lay large amounts of synchronous eggs, or because they give live birth rather than lay eggs (as in the case of viviparous species such as tsetse flies and aphids). These restrictions dramatically limit the use of CRISPR-Cas9 technology across diverse systems.

Most female oviparous animals deliver protein material to their developing ovaries through a conserved process of ovary and egg maturation called vitellogenesis. In insects and other arthropods, yolk protein precursors (YPPs) are synthesized in the fat body, secreted into the hemolymph, and are taken up into the ovaries by receptor-mediated endocytosis (RME)[7–9]. During vitellogenesis, multiple receptors in the oocyte membrane are available and bind YPP ligands that are internalized, accumulated in endosomal vesicles, and sorted into yolk granules for nutrient storage for the developing embryo[10–12].

The use of specific ligands to deliver material into mammalian cells by receptor-mediated endocytosis has been explored for drug delivery since the 1980s[13–15]. For example, when the protein transferrin was used as a ligand and chemically conjugated to molecular cargo such as toxins[16], liposomes[17,18] proteins[13], or DNA[19], these molecules were internalized into the cell via the transferrin receptor and released into the cell cytoplasm in vitro and in vivo[20–22]. Delivery efficacy depends on successful release of cargo from the endosomes and lysosomes[13,17–20,23,24], often by chemical membrane destabilizers such as ammonium chloride, amines, chloroquine, or monensin[13,15,25,26].

We hypothesized that a ligand derived from arthropod YPPs could be fused or bound to molecular cargo such as the Cas9 ribonucleoprotein (RNP) complex (Cas9 complexed with a single-guide RNA (sgRNA)) and, when injected in the hemolymph of vitellogenic females, be delivered into the oocyte at levels necessary to achieve genome editing in the embryo, bypassing the requirement for embryonic microinjection. We term this strategy Receptor-Mediated Ovary Transduction of Cargo (ReMOT Control) (Supplementary Fig. 1). Mosquitoes are excellent models for development and testing of this technology because synchronous egg development can be induced by blood feeding, and significant literature exists on vitellogenesis and receptor-mediated internalization of mosquito yolk proteins[7,27–30]. Here we report the successful delivery of Cas9 RNP to mosquito ovaries and subsequent targeted gene editing of the disease vector mosquito *Aedes aegypti* at efficiencies comparable to traditional embryo microinjection.

## Results

**Delivery of protein cargo into mosquito oocytes using P2C.** *Drosophila melanogaster* Yolk proteins (DmYP) 1, 2, and 3 are small (50 kD) YPPs that are recognized by receptors belonging to the protein superfamily of low-density lipoprotein receptors. This family includes vitellogenin receptors, lipophorin receptors, and yolk protein receptors that are present in oocytes of diverse oviparous animal species and that can recognize related YPP ligands[31]. A previous study[32] used immunohistochemistry to demonstrate that the 439 amino acid (aa) *D. melanogaster* Yolk Protein 1 (DmYP1) was internalized by the oocytes of the mosquito *Anopheles gambiae* when recombinant DmYP1 was injected in the hemolymph of vitellogenic females. To test whether DmYP1 (or derivatives thereof) might act as suitable ligands to deliver protein cargo into mosquito oocytes, a DmYP1 fusion protein containing the enhanced green fluorescent protein (EGFP) was expressed using *D. melanogaster* S2 cells transfected with the plasmid pAc5-DmYPss-DmYP1-EGFP (see Methods). The protein was injected into the thorax of *A. gambiae* females 12 and 24 h post-blood feeding (PBF) (Supplementary Fig. 2). Recombinant EGFP lacking a targeting ligand was injected as a negative control. Ovaries dissected at 24, 36, 48, 60, and 72 h PBF were examined for visible EGFP fluorescence. EGFP was visualized in >98% of primary oocytes regardless of the timing of the injection (Supplementary Table 1) and at all stages of oocyte development including in fully developed eggs ready to be oviposited (Supplementary Fig. 3).

For easier downstream construct design and expression, we used deletion analysis to identify a smaller region of DmYP1 sufficient for uptake into mosquito ovaries (Fig. 1a). Fragments containing the 120 aa N-terminal portion of the protein ("P2") could be taken up into ovaries (Fig. 1b). Further deletion analysis of the P2 fragment identified that a 41 aa fragment, termed P2C (NLQQQRQHGKNGNQDYQDQS-NEQRKNQRTSSEEDYSEEVKN), was fully sufficient to deliver EGFP to *A. gambiae* oocytes (Fig. 1c). To test the functional range of the identified ligand, P2C-EGFP was injected into five additional species of Anophelinae and Culicinae female mosquitoes following a blood meal. EGFP signal was observed in the ovaries of all tested species at uptake efficiencies comparable to analogous tests in *A. gambiae* females (Fig. 2a).

For gene-editing experiments, we focused on the mosquito *A. aegypti* as multiple target genes and sgRNAs have already been validated for this species[33], allowing us to directly compare ReMOT Control efficacy to standard embryonic microinjection-based delivery. We expressed and purified P2C as fusion protein with Cas9 (P2C-Cas9 and P2C-EGFP-Cas9) (Supplementary Fig. 4a, b). After injection in vitellogenic *A. aegypti* females, Cas9, EGFP, or both (depending on the injected protein) were detected by fluorescence/immunofluorescence in the developing oocytes (Fig. 2b).

**Outcross strategy to test gene editing by ReMOT Control.** To test gene-editing efficiency of ReMOT Control, we targeted the kynurenine monooxygenase (*kmo*) gene in *A. aegypti*. When homozygous, the recessive white alleles (*kmo^w*) produce white-eye female mosquitoes (instead of the wild-type dark eyes) that are easily identified immediately after hatching, facilitating the screening of the mutated phenotypes (see larvae, pupa, and adult in Fig. 3a). We validated the activity of two sgRNAs targeting nucleotides 460 (sgRNA460) and 519 (sgRNA519*) of the exon 5 of *kmo* previously identified by Basu et al.[33]. First, to test whether the P2C ligand affected the function of Cas9, fusion proteins Cas9-P2C, Cas9-P2C-EGFP, and unmodified Cas9 (control) were used to target the *A. aegypti kmo* gene using sgRNA460 by standard microinjection into wild-type embryos. All three proteins were active in embryos and generated mutations at the *kmo*-460 site (Supplementary Table 2). Resulting mutant larval phenotypes including mosaic and white-eyed individuals were pooled to generate a multi-allelic colony of white-eye mosquitoes with mutations at the 460 site (Supplementary Table 2, Fig. 3a, b). One individual female was used to find an isoallelic line (Wh-Iso8-kmo^460) with a characterized *kmo* site 460 deletion (del-5′ GGTGATCATT3′). We then outcrossed wild-type mosquitoes

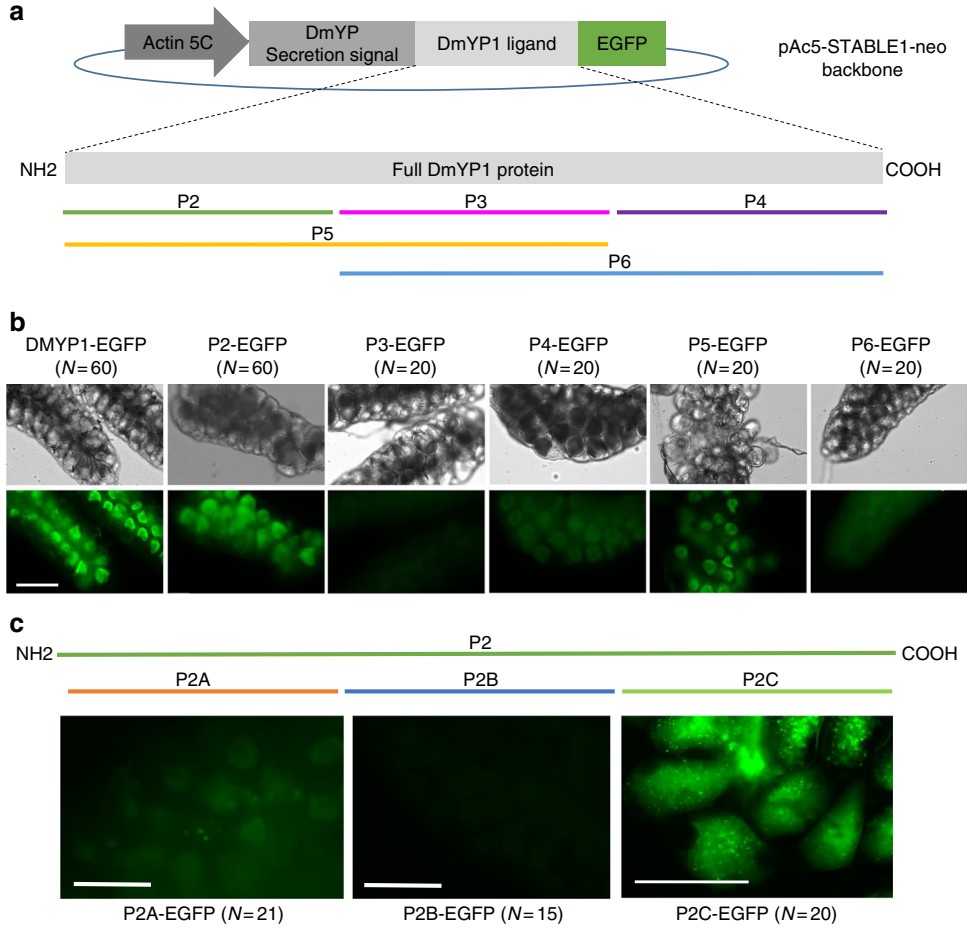

**Fig. 1** Deletion mapping of *D. melanogaster* DmYP1 and P2C-mediated delivery of EGFP. **a** Schematic showing the construct pAc5-STABLE1-neo backbone modified to express and secrete EGFP fused to DmYP1 or any of its derivatives (P2, P3–P4, P5, or P6) under the Actin5C promoter in S2 cells. Fragments P2, P3, and P4 are about 120 amino acids of DmYP1 each. Portion P5 contains portions P2 and P3. Portion P6 contains P3 and P4. **b** EGFP fusion proteins containing DmYP1 or each fragment (P2–P6) as ligand were injected into the hemolymph of vitellogenic *A. gambiae* females. Ovaries were dissected 16 h after injections. Fragments containing the N-terminal region of DmYP1 showed the highest transduction of EGFP into the ovaries. **c** Deletion analysis of the P2 region. EGFP was detected at the highest intensity when fused to the P2C portion of DmYP1. Scale bar is 100 μm

(genotype $kmo^{460+519+}/kmo^{460+519+}$) male to male individuals of this line ($kmo^{460w519+}/kmo^{460w519+}$) and injected the out-crossed females with P2C-RNPs complexed with sgRNA519*. In the absence of editing, all offspring from these crosses are het-erozygous for the 460-site mutation ($kmo^{460w519+}/kmo^{460+519+}$) and have black eyes (Fig. 4a, b). If editing of the 519 site was successful ($kmo^{460w519+}/kmo^{460+519w}$), the cross would result in white-eye offspring since mutations at the *kmo* sites 460 and 519 are complementary (Fig. 4c). This strategy allowed us to screen for white-eye individuals carrying mutations at the 519 site induced by RNP delivered via ReMOT Control (see Supplementary Methods for details about processing the females after injection and offspring screening).

**Optimization of injection conditions.** To test for editing of the oocyte chromosomal sequence, we injected wild-type females outcrossed to white-eye males (Wh-Iso8-kmo$^{460}$ in most cases; the multi-allelic white-eye line was used for early experiments) with Cas9 RNPs using the sgRNA519* (Fig. 4a). Injections without an endosomal release reagent (EER) did not result in editing events in these experiments (Supplementary Tables 3, 6). We observed a focal pattern of EGFP and Cas9 fluorescence in ovaries following injection with P2C-EGFP-Cas9 (Fig. 2) and the body of literature surrounding drug delivery efforts supports the

confinement of cargo to endosomes following RME[13,20,24–26,34]. We therefore sought to explore whether EERs characterized in these drug studies would promote editing of the germline DNA by mediating the release of the P2C-EGFP-Cas9 from endosomes. We tested the effect of ammonium chloride, chloroquine, sapo-nin, and monensin on ReMOT Control efficiency. A set of pilot injections allowed us to identify an estimated LD$_{50}$, which was used for injections; later as injection technique improved, we noticed that this concentration resulted in higher survival rates. Using the protein P2C-EGFP-Cas9 (to track protein entry into ovaries), we conducted troubleshooting experiments to identify optimal conditions for gene editing. We detected gene-editing events with P2C-EGFP-Cas9 concentrations between 2200 and 6700 ng μL$^{-1}$, sgRNA concentrations >1000 ng μL$^{-1}$, and the EER chloroquine at a concentration between 0.5 and 2 mM and injecting females ≥24 h post-blood meal (Supplementary Table 4). These concentrations are higher than is necessary for embryo microinjections, but in our hands, female survival was generally between 60% and 80% (Supplementary Tables 4, 5) and provided sufficient offspring (>20 viable offspring/female on average) to detect mutants.

**Gene editing of the maternal allele.** We used these optimized conditions to measure the ReMOT Control editing efficiency of

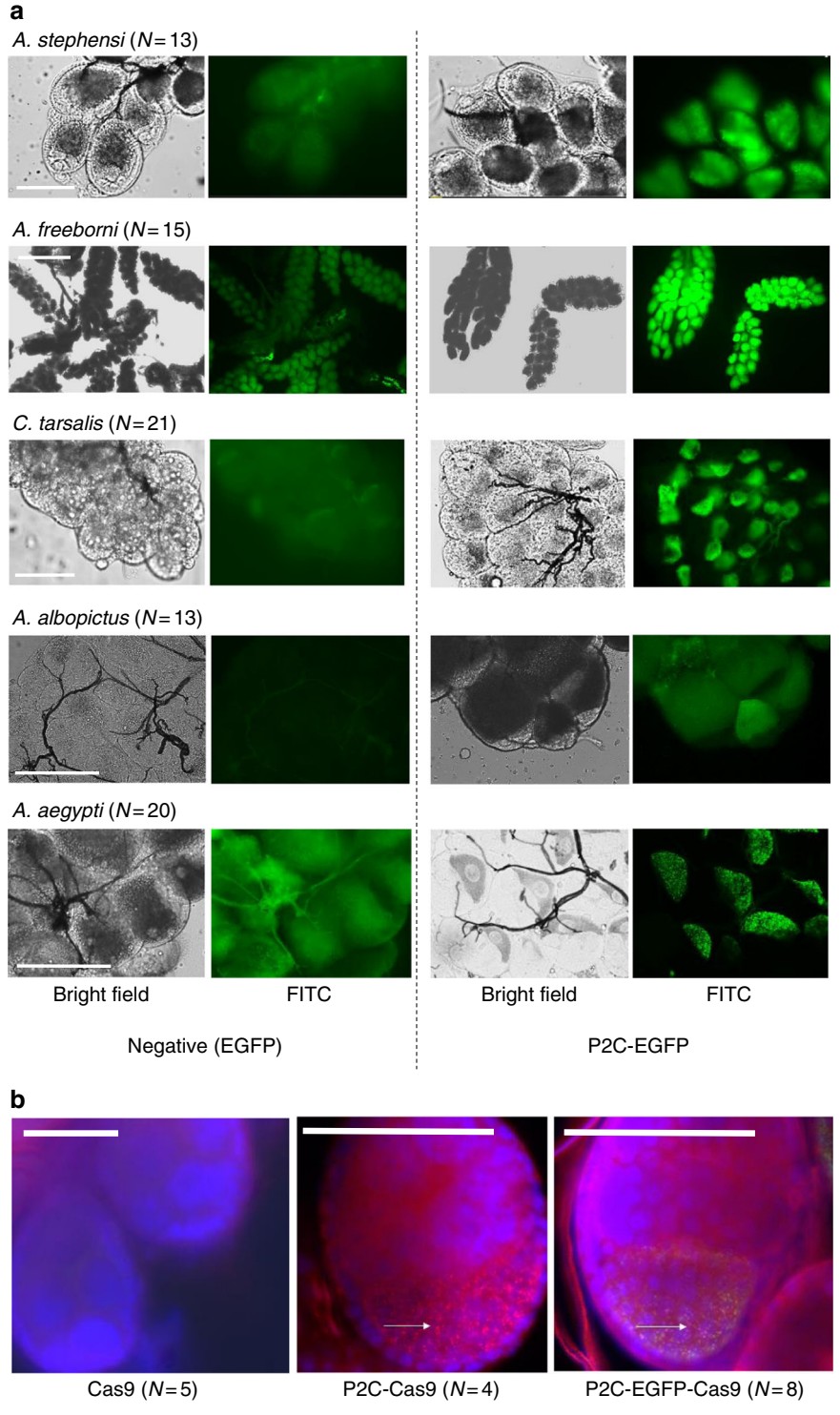

**Fig. 2** P2C-mediated protein translocation in several mosquito species. **a** P2C-EGFP, expressed in *E. coli* using the plasmid pET28a (See details in Supplementary Fig. 3), was injected into the hemolymph of several species of vitellogenic mosquitoes. Negative control injections were performed with unmodified EGFP. Scale bars: *A. stephensi*—50 μm; *A. freeborni*—500 μm; *C. tarsalis*, *A. albopictus*, *A. aegypti*—100 μm. **b** Merged images of fluorescence detected in immunofluorescence assays on ovaries of *A. aegypti* females injected intrathoracically with different P2C-Cas9 proteins. P2C-Cas9 and P2C-EGFP-Cas9 showed distinctive granules (white arrows) containing Cas9-; these granules are co-localized with EGFP in P2C-EGFP-Cas9. From left to right: Cas9 (No ligand), P2C-Cas9, P2C-EGFP-Cas9. Blue = DAPI, Red = Cas9 antibody, Green = EGFP, Yellow = overlap in red and green signal. Scale bars = 50 μm

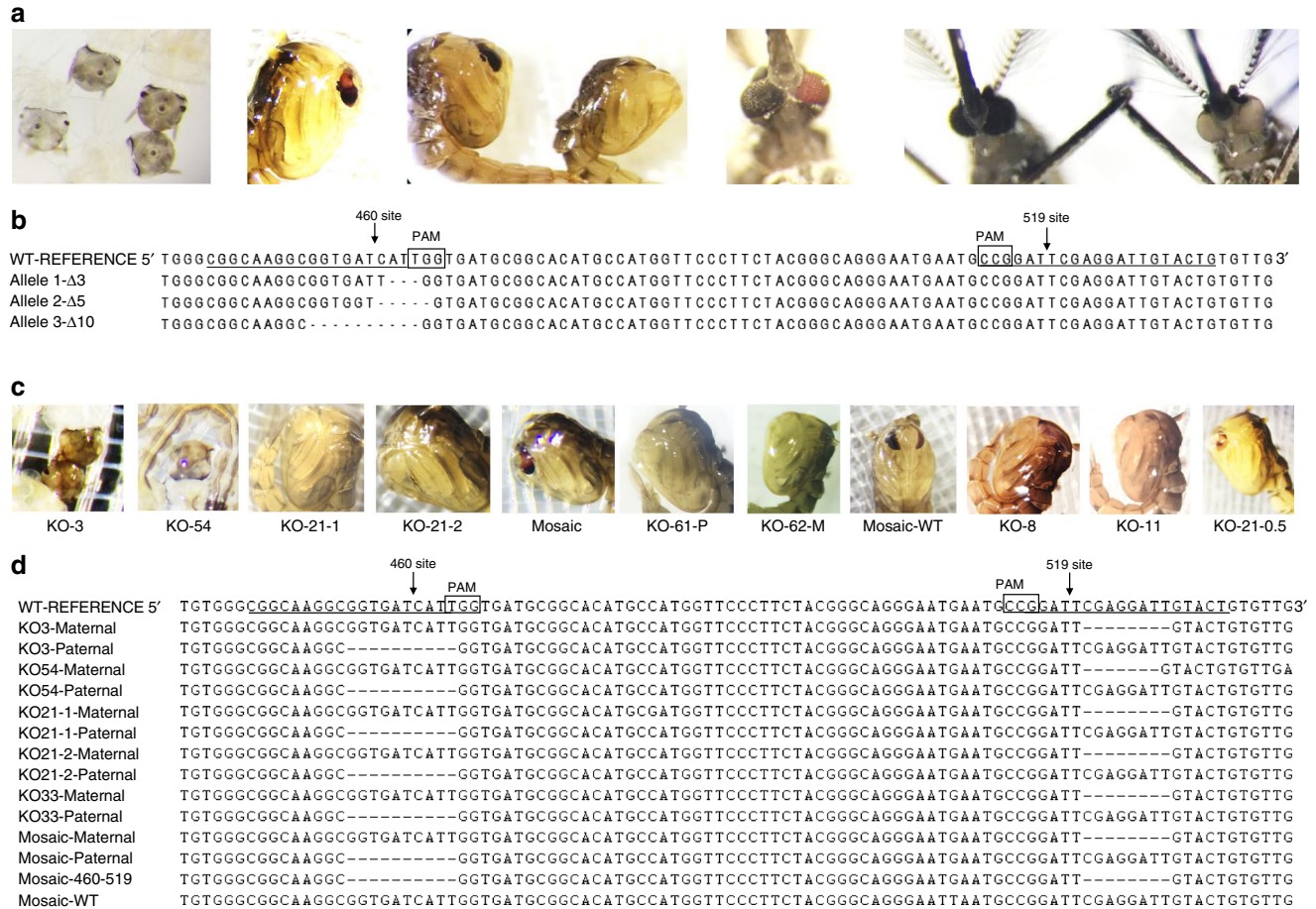

**Fig. 3** Cas9-mediated gene editing in *A. aegypti* by embryo injection and ReMOT Control. **a** Example phenotypes of individuals obtained after standard embryonic microinjection into *A. aegypti* embryos of Cas9, P2C-Cas9, and P2C-EGFP-Cas9 fusion RNP complexed with sgRNA460. From left to right: $G_0$ white and wild-type larvae; mosaic pupae; wild-type and white pupae; mosaic adult; wild-type and white adults. **b** Genotypes of alleles obtained at the 460 site from selected recovered white-eye offspring. The allele 3Δ-10 was isolated in a colony of white-eye mosquitoes (Wh-iso8-kmo[460]). **c** Phenotypes of offspring obtained after female injection of ReMOT Control RNPs complexed with sgRNA519* into *A. aegypti* females. **d** Genotypes of obtained modified offspring. Note the maternal and paternal allele clone sequences of knockout individuals. Mosaic individuals showed both mutated sites and intact WT sequences in clones from the same mosquito

P2C-EGFP-Cas9, P2C-Cas9, and unmodified Cas9 as a control. We calculated efficiency of ReMOT Control-mediated maternal allele editing using two measures in an attempt to compare gene-editing efficiencies to embryo injections: (i) effort efficiency (EEF; the proportion of edited $G_0$ out of the number of individuals injected [embryo or females]), and (ii) $G_0$ gene-editing efficiency (GEF; the proportion of edited $G_0$ out of the total number of larvae hatched).

We injected unmodified Cas9 RNP (negative control) into a total of 138 females (at either 24 or 54 h post-blood meal). Across all experiments, we observed a single edited offspring out of the 2617 hatched larvae (EEF: 0.7%, GEF: 0.04%). In comparison, the addition of the P2C ligand to Cas9 increased EEF up to >30% (approximately 1 mutant per ~3 females injected) and GEF to >1.5% (42.9- and 38-fold respective increase compared to native Cas9). The efficiency for P2C-EGFP-Cas9 was lower (EEF: 4.9% and GE: 0.61%) than P2C-Cas9 but higher than Cas9 alone (5 and 9 times, respectively) (Table 1). Larger proteins that included the EGFP marker had lower editing efficiency compared to P2C-Cas9 whether injected into females or embryos (Table 1, Supplementary 2). We observed that, for both P2C-Cas9 and P2C-EGFP-Cas9 injections, all mosaics were produced from females injected at later time points post-blood meal; edited individuals emerging from females injected at earlier time points had white-eye

phenotype, rather than mosaic (Table 1, Fig. 5). If this observation represents a biological difference in access to the $G_0$ DNA, it may be important to consider timing in future applications of ReMOT Control in mosquitoes.

**Gene editing of paternal alleles by ReMOT Control.** In order to estimate the frequency for gene editing of the paternal allele, homozygous Wh-Iso8-kmo[460] females outcrossed to wild-type males were injected with P2C-Cas9 or P2C-EGFP-Cas9 at 54 h PBF. In preliminary experiments, we detected no altered offspring from treatments with chloroquine or ammonium chloride. We produced one mosaic offspring using saponin as the EER and one mosaic offspring using monensin as the EER. Additionally, we detected one white-eye offspring from an injection without any EER (Supplementary Table 7), suggesting that under rare circumstances exit from the endosome may happen without the action of an EER. When we injected P2C-Cas9 with chloroquine as the EER, editing of the paternal allele was highly efficient (EEF: 33.3% and GE: 0.7%) (Table 1).

**Simultaneous gene editing of paternal and maternal alleles.** An experiment was conducted to validate simultaneous Cas9-mediated editing of both the maternal and paternal alleles. Wild-type intercrossed females were injected with P2C-EGFP-Cas9 at 54 h

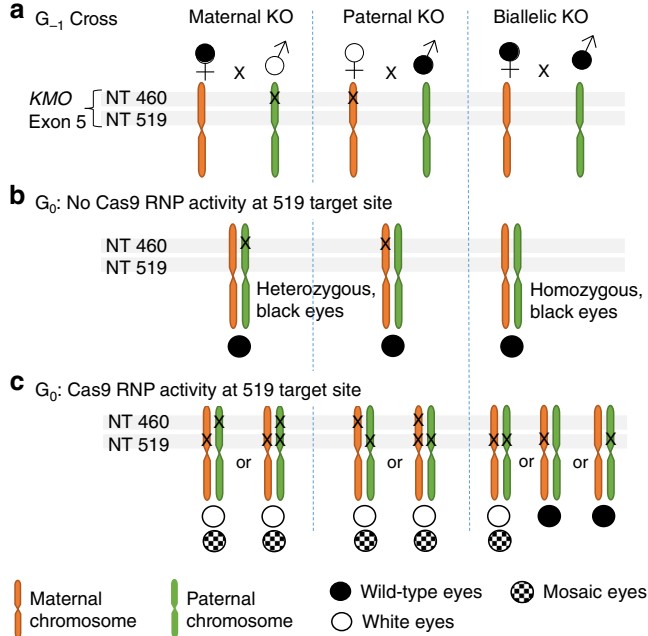

**Fig. 4** Crossing scheme for distinguishing maternal and paternal *kmo* allele editing. **a** Homozygous, white-eye female or male mosquitoes (open white symbols) with a deletion on the *kmo* gene exon 5 at nucleotide (NT) 460 were outcrossed to wild-type mosquitoes (black symbol) to identify the effects of Cas9 activity in single chromosomes (paternal or maternal). Wild type intercrosses were used to test for simultaneous gene editing in the paternal and maternal chromosomes. An "x" on the illustrated chromosome indicates mutation at that site. **b** Genotype and phenotype resulting from the $G_0$ crosses in the absence of Cas9 activity; there is no mutation on the 519 site, the progeny are either heterozygous for mutation at 460 or wild-type homozygous, and therefore the phenotype of individuals is wild type. **c** Genotype and phenotype resulting from the $G_0$ crosses in the presence of Cas9 activity at the 519 site. Progeny from intercrosses are white or mosaic eyes. Monoallelic activity at the 519 site is undetectable in $G_0$ progeny from the $G_{-1}$ wild-type intercross, but biallelic activity is detectable as white eyes or mosaics. Further crossing of $G_0$ can lead to detection of monoallelic activity

PBF using the optimal concentrations of reagents determined in previous experiments and pooled for oviposition. We screened 1413 larvae and found one mosaic $G_0$ individual (out of the 40 surviving females, see Supplementary Methods for details on treatment of pooled females for oviposition) (Table 1). This individual died as a pupa. Remaining $G_0$ wild-type males and females were pooled by sex, outcrossed to Wh-Iso8-kmo$^{460}$ adults of the appropriate sex, and $G_1$ progeny from each outcross was screened for white eyes. We found nine white-eye larvae out of the 11,161 larvae from these outcrosses.

**Sequence verification of gene-editing events**. To verify knockout events, the sequence flanking both target sites (460 and 519) was amplified from genomic DNA of recovered white-eye and mosaic offspring (Fig. 3c), cloned, and sequenced. Sequences from at least six clones per individual were obtained to confirm maternal vs paternal gene-deletion events. The sequences of the *kmo* gene of all white individuals showed site 460 and 519 indels on separate clones (Fig. 3d), confirming ReMOT Control gene editing took place and indicating that the maternal chromosome was targeted. Multiple alleles at the 519 site including wild-type sequences and indels at both sites on the same clone were detected in clones derived from mosaic individuals, indicating that the maternal and paternal chromosomes were targeted.

**Heritability of the *kmo* mutations**. A sample of $G_0$ individuals showing white eyes (1 male and 2 females from outcrossed, injected wild-type female generation −1 ($G_{-1}$)) and mosaic (1 male and 1 female from outcrossed, injected white-eye $G_{-1}$ female) phenotypes were outcrossed to homozygous Wh-Iso8-kmo$^{460}$ mosquitoes and the progeny screened for white eyes. We observed that 100% of the progeny resulting from white-eye $G_0$ females had white eyes. Sequencing of *kmo* PCR fragments showed both the paternal 460 site mutation and the maternal site 519 mutation in $G_0$ and $G_1$ individuals with white eye-phenotype. On the other hand, we were unable to confirm heritable gene editing in the $G_0$ offspring from the two mosaics since ~50% had white eyes, consistent with what we would expect from crossing a heterozygous $G_1$ with white-eye mosquitoes (Supplementary Table 8).

## Discussion

We have developed a method for arthropod gene editing that circumvents the need for embryo injections and has potential for broad application to systems recalcitrant to traditional embryonic microinjection techniques. By exploiting the endogenous YPP uptake pathway that is conserved across oviparous animals, we have targeted molecular cargo to the oocytes of six species of mosquitoes and have engineered the transport of Cas9 to the germline for gene editing in *A. aegypti*. Mosquitoes are good model organisms in which to develop genetic technologies, especially *A. aegypti*, which are easy to rear and are straight forward to manipulate genetically. This species has been the initial target of many of the genetic technologies developed for mosquitoes thus far, including the initial transformation of mosquitoes using transposons[35,36], the application of Cas9-mediated gene-editing[5,33,37], and most recently the development of a Cas9-expressing transgenic line for highly efficient gene editing, which allows heritable gene disruption at consistently high efficiencies by embryo microinjection[38].

Using this species, we developed a method to deliver Cas9 by adult injection to the germline tissue for targeted disruption of a visual marker. A 41 aa peptide (P2C) derived from DmYP1 was sufficient to mediate Cas9 uptake into the ovaries, and when coupled with an appropriate EER, the Cas9 RNP specifically targeted the *A. aegypti kmo* gene in a manner that is both heritable and flexible in terms of timing and efficiency. The phenotypes we observed following injection of the P2C-Cas9/sgRNA RNP into adult females likely reflect the developmental timing of the mosquito oocyte and the embryo, suggesting that accessibility of the chromosomal DNA to the RNP in the oocyte changes during development of the ovaries (Fig. 5). That paternal copies of the gene are mutated suggests that the complex is stable and active following fertilization, which is >24 h after injection[39]. Earlier injections (~24 h PBF) produced maternal chromosome-editing events at high efficiency, while mosaic $G_0$'s were observed from later injections PBF, indicating editing events in some somatic cells, and confirming that the RNP complex is active in the fertilized embryo.

We detected multiple editing events at the 519 target site that when sequenced revealed the same deletion (Fig. 3d). The double-stranded break cause by Cas9 can be repaired using non-homologous end joining (NHEJ) or by homology-mediated mechanisms, either homology-directed repair (HDR), if a repair template is available, or by microhomology present on either site of the target site (microhomology-mediated end joining) first evidenced in budding yeast[40–42], recently described to have important impacts on Cas9-editing outcomes[43]. Deletions generated by NHEJ repair of a cut site would randomly result in one third of deletions being in-frame, whereas microhomology could

**Table 1 ReMOT Control gene editing using optimized concentrations for EER and the RNPs Cas9, P2C-Cas9 and P2C-EGFP-Cas9s**

| Protein | Time of injection PBF | Individual injections (N) | Females laying KO | Females laying KO (%) | G₀ larvae hatched (N) | G₀ mosaic (N) | G₀ white (N) | #KO/ injection EEF (%) | #KO/ surviving G₀ GEF (%) |
|---|---|---|---|---|---|---|---|---|---|
| Maternal gene editing: wild-type females × white-eye males (mono-allelic) | | | | | | | | | |
| Cas9 | 24 h | 83 | 1 | 1.20% | 1527 | 0 | 1 | 1.2% | 0.1% |
| | 54 h | 55 | 0 | 0% | 1090 | 0 | 0 | 0.0% | 0.0% |
| | Total | 138 | 1 | 0.70% | 2617 | 0 | 1 | 0.7% | 0.0% |
| P2C-Cas9 | 24 h | 88 | 2ᵃ | 2.20% | 1798 | 0 | 18 | 20.5% | 1.0% |
| | 54 h | 54 | 9 | 16.60% | 1052 | 2 | 23 | 46.3% | 2.4% |
| | Total | 142 | 11 | 8% | 2850 | 2 | 41 | 30.3% | 1.5% |
| P2C-EGFP-Cas9 | 24 h | 90 | 3 | 3.00% | 534 | 0 | 6 | 4.4% | 1.1% |
| | 54 h | 134 | 5 | 2.90% | 1261 | 3 | 2 | 3.7% | 0.4% |
| | Total | 224 | 8 | 4% | 1795 | 3 | 8 | 4.9% | 0.6% |
| Paternal gene editing: white eye (or heterozygous females) × wild-type males (mono and biallelic) | | | | | | | | | |
| P2C-Cas9 | 54 h | 12 | 3 | 25% | 557 | 0 | 4 | 33.3% | 0.7% |
| P2C-EGFP-Cas9 | 54 hᵇ | 18 | 1 | 5.60% | 9 | 0 | 1 | 5.6% | 11.1% |
| | 54 h | 40 | 2 | 5.00% | 213 | 2 | 0 | 5.0% | 0.9% |
| | Total | 58 | 3 | 5.20% | 222 | 2 | 1 | 5.2% | 1.4% |
| Maternal and paternal gene editing: wild-type females × wild-type males (biallelic) | | | | | | | | | |
| P2C-EGFP-Cas9 | 54 h | 58 | 1 | 1.70% | 1413 | 1 | 0 | 1.7% | 0.1% |

ᵃAt 24 h injections, two isofemales produced only 7 out of the 18 knockout larvae, and the number of females producing the other 11 knockouts is not known because those females were mass-reared
ᵇThe only trial of P2C-EGFP-Cas9 injected without an EER that produced one knockout

increase the abundance of in-frame repair up to 100%. An analysis of 40 bp on either side of the 519 target site using a microhomology predictor tool (http://www.rgenome.net/michcalculator/[43]) revealed that the first and third most common repairs would be in-frame (6- and 18-bp deletions, respectively) and that the second most common deletion mutant is the one that we consistently observed in our white-eye knockouts. The out-of-frame score for this target site, 61%, is only slightly smaller than the recommended 66% to match randomness by NHEJ, so it is not likely that microhomology decreased the detection of mutants at the 519 target site, but these data highlight the importance of target-site analysis prior to use, as microhomology can substantially prohibit or improve the generation of loss-of-function deletions and detection of mutants.

In further application of this method, researchers will desire mutations in genes with no visible phenotype. Until ReMOT Control is modified to deliver donor DNA containing a visual marker gene to the germline tissue for integration by HDR, mutants will have to be detected using molecular PCR-based methods, such as T7 assays, high-resolution melting assay, or sequencing of amplicons from the target gene. These methods have already been demonstrated as effective for detecting mutants following embryo microinjection[5,33,44–47]. The mutation rates detected in individuals heterozygous for *kmo* disruption reported here using ReMOT Control were lower than what was reported for the same site following embryo injection[33] so mutant detection will require screening a higher number of G₀ offspring. It is possible that a combination of new techniques like germline Cas9 expression and ReMOT Control may allow robust, consistent, and technically straightforward genetic manipulation of mosquitoes. Insertion of a marker gene into target sites would substantially improve screening; for ReMOT Control this will require the delivery of donor DNA to the germline of injected female mosquitoes. Previous studies have shown that delivery of DNA to the ovary is possible, but successful efforts to establish transgenesis by adult injection have not been reported[48,49]. Efforts are

underway to apply ReMOT Control to HDR-mediated transgenesis following adult injection of the P2C-Cas9/sgRNA complex.

DmYP1 or its derivatives, including P2C, are not homologous to any known vitellogenin sequences either in arthropods or vertebrates. However, DmYP1 shows homology with several protein sequences from lipases in vertebrates and yolk proteins in cyclorrhaphan dipterans[50,51]. Further investigation is underway to identify the specific receptor that mediates ovary-specific uptake in *A. aegypti*. Application of ReMOT Control to other species will require a consideration of system-specific factors to optimize efficiency. ReMOT control could theoretically be extended to any species that uses a receptor and a ligand to transduce material in their oocytes. One crucial factor is to use the appropriate ligand/receptor pair for the species of interest to deliver Cas9 or other protein cargo into the oocytes. A second relevant factor is time of injections before oviposition. We demonstrated that the RNP remains active in the oocyte for a limited time; therefore, choosing the right time for injection may increase efficiency. Optimizing injection component conditions such as amount of RNP injected and type and amount of endosomal escape reagent used will be critical to success using ReMOT Control in non-model species.

Although we validated ReMOT Control in mosquitoes, the technique can be conceptually extended to any animal species that undergoes vitellogenesis (most invertebrates and non-mammalian vertebrates). The P2C ligand or others derived from DmYPs may work generally in dipterans (although this must be formally tested), as YP receptors have been shown to uptake YPPs from 13 other *Drosophila* species and 5 non-drosophilid dipterans[52]. DmYPs also share sequence homology with two minor YPPs of lepidopterans, the egg-specific protein and the follicular epithelium yolk protein[53,52].

Compared to embryo injection, gene editing by ReMOT Control was efficient and technically much easier to accomplish. The requirements for ReMOT Control per G₀ targeted for

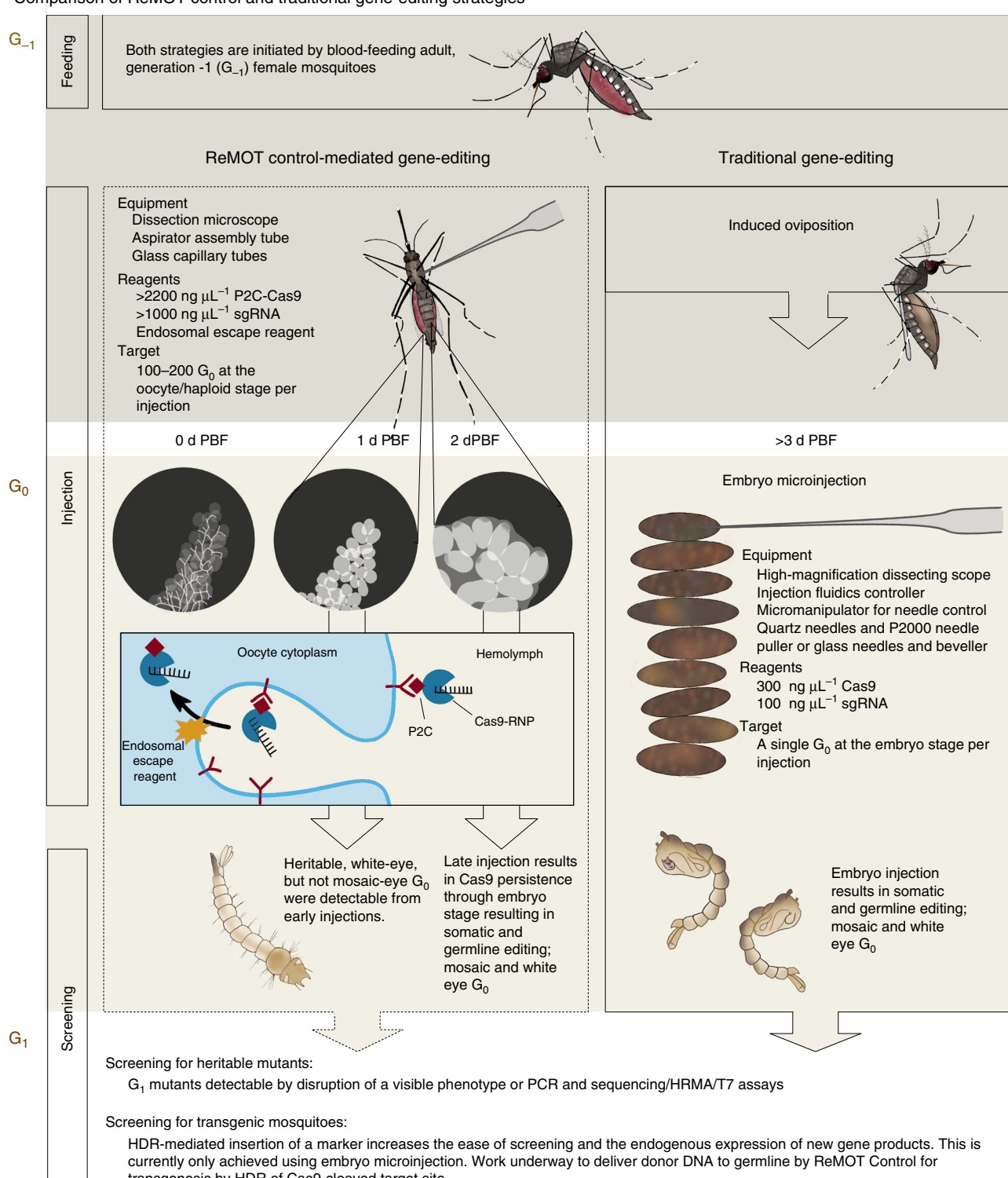

**Fig. 5** Schematic description and model of ReMOT Control vs embryo injection for Cas9-mediated gene targeting. ReMOT Control-mediated and traditional gene-editing strategies are compared in terms of methods (injection stage and timing, equipment, reagents, screening). Generation designations (Gx) are indicated in brown letters at the top left of each panel. A schematic model of the uptake of P2C-Cas9 RNP into the mosquito oocyte is illustrated in the $G_0$, ReMOT Control panel. PBF post-blood feeding, RNP ribonucleoprotein, sgRNA single guide RNA, HRMA high-resolution melting assay, HDR homology-directed repair

mutation are substantially lower compared to embryonic micro-injection where the microinjection apparatus can cost thousands of dollars and require extensive training to use. In contrast, the equipment for ReMOT Control injections costs approximately 2 dollars, a standard (non-laser) capillary puller or a flame can be used to pull glass needles or affordable pre-pulled glass needles can be purchased from commercial vendors (in the absence of a needle puller), and the technique can be learned in less than an hour. While higher concentrations of Cas9 protein and sgRNAs are required in the injections mixes, the cost to produce these reagents is substantially lower than traditional microinjection equipment. Further, one injection with these higher concentrations potentially targets hundreds of $G_0$ offspring within the injected $G_{-1}$ female. Along with the financial improvements, the ease of adult injections makes this method a substantial improvement over existing embryo-injection techniques, putting gene-editing capability into the reach of non-specialist laboratories and non-model systems and potentially revolutionizing the broad application of functional arthropod genetics.

## Methods

**Mosquitoes.** *A. gambiae* (Keele strain), *A. stephensi* (Liston strain), *A. freeborni* (F1 strain), *Culex tarsalis* (Yolo strain), *Aedes albopictus* (Houston strain) and *A. aegypti* (Rock strain) were reared at 27 °C, 75 ± 10% relative humidity, 12 h light: 12 h dark in a walk-in environmental chamber. Larvae were fed with ground Koi pellets or Tetramin daily. Adults were provided ad lib with 10% sterilized sugar on a cotton wick. For injection experiments female mosquitoes were fed on expired anonymous human blood (Biospecialty Corp.) using a water-jacketed membrane feeding system. Blood-fed females from the same cage were divided evenly and randomly into experimental groups prior to injection.

**Female injections and dissections.** Injections were performed with an aspirator tube assembly (A5177, Sigma) fitted with a glass capillary needle. Adult females were immobilized at 4 °C following a bloodmeal and kept on ice during injection. Females were injected intrathoracically until no additional fluid would enter, at an approximate volume of 200 nL per female. To visually confirm EGFP in the ovaries following injection, ovaries were dissected, mounted in saline buffer mixed with SlowFade Gold® antifade reagent (Invitrogen), covered with a coverslip, and imaged on an Olympus BX41 epifluorescent microscope. Negative controls were injected with recombinant EGFP lacking a targeting ligand. Samples were not blinded for analysis.

**Identification of the DmYP1 mosquito receptor-binding region.** A deletion analysis of the DmYP1 protein was conducted to identify the smaller region that efficiently transduced cargo into mosquito ovaries. Three portions of approximately 120 aa (Fig. 2a) were encoded on an expression plasmid linked to EGFP via a Glycine-Serine linker (see Supplementary Notes for DNA sequences of all fusion proteins). The construct was inserted into the plasmid pAc5-STABLE1-neo[54] under control of the *Drosophila* Actin5C promoter. Plasmids were transfected into *Drosophila* S2 cells using Lipofectamine® LTX WITH Plus™ Reagent (Thermofisher) following the manufacturer's protocol. Cell culture supernatant was collected 72 h post-transfection, concentrated 100× using Amicon® and Centricon® Plus-70 centrifugal filter devices (Millipore), and injected into female *A. gambiae* 12 and 24 h PBF. Ovaries were dissected 12, 24, 48, and 72 h postinjection and visualized for EGFP fluorescence. EGFP fusion protein fused to the full-length DmYP1 protein (439 aa) was used as a positive control, while recombinant EGFP without a targeting ligand was used as a negative control. To identify fragment P2C, fragment P2 was split into 3 approximately 40 aa fragments, fused to EGFP, and proteins expressed and purified in *E. coli* as described for Cas9 proteins below.

**Fusion Cas9 protein expression and purification.** The DmYP1 fragment P2C and EGFP were cloned as a Cas9 fusion into the plasmid pET28a-Cas9-cys (Addgene #53261) to create the constructs pET28a-P2C-Cas9 and pET28a-P2C-EGFP-Cas9. Plasmids were transformed into *E. coli* BL21 (NEB) and Rosetta2™ (DE3) pLysS competent cells Novagen (Millipore Sigma) and transformation verified by PCR. To induce the expression of recombinant protein, a preculture was grown overnight at 30 °C in 50 mL of Luria Broth (LB) supplemented with kanamycin for BL21 or chloramphenicol and kanamycin for Rosetta2 (35 and 100 μg μL⁻¹, respectively). After 12 h, 10 mL of preculture was transferred to 990 mL of LB supplemented with the same antibiotics, incubated at 37 °C until OD = 0.6, when 0.05 mM Isopropyl β-D-1-thiogalactopyranoside was added and the culture incubated overnight at 30 °C.

Cells were spun down at >10,000 × *g* for 15 min, resuspended in 50 mL lysis buffer (20 mM Tris-HCl pH 8.0, 300 mM NaCl, 20 mM imidazole), placed at −80 °C overnight, and incubated with lysozyme (1 mg ml⁻¹) and 100 ppm of

parametHylsulfoxide (PMSF) for 30 min at 4 °C. The suspension was sonicated 5 times at 60% duty 5 s pulse 5 s rest (two aliquots each 25 mL), centrifuged at 13,000 × *g* for 30 min, and the supernatant removed and incubated with Ni-NTA beads (Qiagen) with rotation at 4 °C overnight. Beads were washed 3 times with 10 mL lysis buffer and eluted with 1 mL elution buffer (20 mM Tris-HCl pH 8.0, 300 mM NaCl, 250 mM imidazole) 10 times. Eluted protein was dialyzed in a Slide-A-Lyzer dialysis cassette (Thermo Fisher) for 2 h in dialysis buffer (50 mM Tris-HCl pH 8.0, 300 mM KCl, 0.1 mM EDTA, 0.5 mM PMSF) and buffer changed every 2 h 2 times, then left overnight at 4 °C with gentle agitation in fresh buffer. Protein purity was visualized by sodium dodecyl sulfate-polyacrylamide gel electrophoresis and concentration estimated using Bradford assay.

**In vitro DNA cleavage assays.** Cas9 fusion proteins and sgRNAs were mixed at a molar ratio of 1:2 and used to test in vitro cleavage activity of a PCR fragment spanning the *A. aegypti kmo*-460 and *kmo*-519 target sites following the PNA BIO suggested protocol (http://www.pnabio.com/products/CRISPR_Cas9.htm). Reactions were performed for 1 h at 37 °C and diagnostic bands visualized by electrophoresis in a 1% agarose/TAE gel (Supplementary Fig. 4c, d).

**Embryonic microinjections.** Embryo microinjection mixes contained Cas9 and Cas9 fusion proteins generated in our laboratory at 300 ng μL⁻¹, and 100 ng μL⁻¹ sgRNA against *kmo*-460. sgRNA was prepared according to the protocol described in Kistler et al.[55] using CRISPR R: 5′AAA-AGC-ACC-GAC-TCG-GTG-CCA-CTT-TTT-CAA-GTT-GAT-AAC-GGA-CTA-GCC-TTA-TTT-TAA-CTT-GCT-ATT-TCT-AGC-TCT-AAA-AC-3′. Briefly, template for in vitro transcription was generated by amplification of the CRISPR_R and target-specific CRISPR_F in 4–8 PCR reactions using Phusion (New England Biolabs). The resulting oligos were pooled from all reactions and purified using the Cycle Pure Kits (Bio-tek). A microgram of PCR product was used in the T7 Megascript kit (Ambion) following the manufacturer's protocol to generate single-stranded sgRNA, which was purified using the Megaclear Column Purification Kit following the manufacturers' protocol. sgRNA concentration was measured using a Nanodrop ND-1000, aliquoted at 2 μg μL⁻¹ and stored at −80 °C.

Protein and the sgRNAs were mixed and incubated at room temperature (RT) for 15 min before injection into *A. aegypti* eggs 90–120 min post-oviposition as described in Jasinskiene et al.[3]. Briefly, eggs were isolated, immobilized on double-sided tape, kept moist with buffer, and injected into the posterior pole. After injection, eggs were placed on wet paper filter for 4 days before hatching. Hatched first-instar $G_0$ larvae were screened for white or mosaic eyes, which were initially pooled to create a white-eye colony. Individuals from this colony were later outcrossed to create an isoallelic white-eye line homozygous for a single $kmo^{460}$ mutation (Wh-Iso8-kmo⁴⁶⁰).

**Immunofluorescence assays.** For immunofluorescence assays, dissected mosquito ovaries were dissected into phosphate-buffered saline (PBS) and fixed with 4% paraformaldehyde for 20 min at RT. To block non-specific binding, ovaries were incubated for 2 h at RT with 3% bovine serum albumen in PBS with 0.1% Tween 20. Following blocking, Cas9 protein was detected using rabbit anti-Cas9 polyclonal antibody (Abcam ab204448) diluted 1:500 in PBS with 0.1% Tween 20 and incubated for 1 h at RT or overnight at 4 °C. After incubation, samples were washed three times with 0.1% Tween 20 in PBS, and the primary antibody labeled with 1:500 anti-rabbit Alexa Fluor 594 secondary antibody (Abcam ab150076). After three additional washes, ovaries were air-dried and slide mounted with ProLong Gold Antifade Reagent (Molecular Probes) and visualized using epifluorescent microscopy.

**Detection of Cas9-mediated editing using ReMOT Control.** After mating, females ($G_{-1}$) were bloodfed and injected with Cas9 (or Cas9 fusion protein) RNPs and EERs. Resulting $G_0$ offspring were screened for *kmo* gene-editing phenotypes (white or mosaic eyes). Genomic DNA from identified individuals was extracted using the Dneasy Blood And Tissue Kits (Qiagen), and the region spanning both *kmo* 460 and 519 sites amplified by PCR using primers ZA2210 (5′TTC-AAG-ACC-AGG-CCT-CAA-TC3′) and KmR1 (5′TCA-CTA-AAC-TCA-GCC-AGT-ATC-CTA-T3′)[33] and cloned into the pJET1.2 vector. A minimum of six clones per individual were randomly selected and sequenced.

**Data availability.** Sequence data for proteins P2C-EGFP-Cas9, P2C-Cas9, and DmYP1-EGFP and its derivatives P2-EGFP, P3-EGFP, P4-EGFP, P5-EGFP, P6-EGFP, P2A-EGFP, P2B-EGFP, and P2C-EGFP are provided in Supplementary material; full sequence of the plasmid vectors are available from the corresponding author upon request. The rest of data supporting the findings of this study are available within the paper and its Supplementary Information files and from the authors upon request.

**Disclaimer.** The paper contents are solely the responsibility of the authors and do not necessarily represent the official views of the Centers for Disease Control and Prevention or the Department of Health and Human Services.

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

## Acknowledgements

This research was supported by NIH/NIAID grants R01AI128201, R21AI111175, R01AI116636, and U19AI089672; NSF/BIO grant 1645331; USDA/NIFA grant 2014-10320; and a grant with the Pennsylvania Department of Health using Tobacco Settlement Funds to J.L.R. D.C.-R. was partially supported by a Fulbright Fellowship and by

Colciencias. V.M.M. was partially supported by USDA/NIFA grant 2017-67012-26101. G.L.H. was supported by NIH grants R21AI124452 and R21AI129507, a University of Texas Rising Star award, the John S. Dunn Foundation Collaborative Research Award, the Robert J. Kleberg, Jr. and Helen C. Kleberg Foundation, and the Centers for Disease Control and Prevention (Cooperative Agreement Number U01CK000512). We thank Chaz Bunce, Erona Ibroci, and Rhiannon Barry for laboratory assistance.

## Author contributions

G.L.H. and J.L.R initially conceptualized and designed the project. D.C.-R., V.M.M, G.L. H., S.P., Y.S., D.K., and D.R.P. conducted the research. D.C.-R., V.M.M., G.L.H., S.P., D. K., and J.L.R. analyzed the data. S.M. and V.M.M. designed and illustrated Fig. 1. D.C.-R, V.M.M., G.L.H., and J.L.R. wrote and edited the manuscript.

## Additional information

**Competing interests:** J.L.R., G.L.H., and D.C.-R. have filed for provisional patent protection on the ReMOT Control technology. The other authors declare no competing interests.

