## [Peer Review File · Nature Communications]

Reviewer Comments:

Reviewer #1 (Remarks to the Author):

This is a very well-done research article describing a new innovative technology for generating heritable germline mutants using receptor-mediated ovary transduction of cargo (ReMOT Control) to deliver the cargo (Cas9 +sgRNA) into the germline by injecting into adult females. Importantly, injecting into adult females overcomes the significant technical barrier of injecting into eggs – a feat that is not possible in many insects/animals. The technology is demonstrated to work in 6 different mosquito species, with the majority of the downstream work focused on *Ae. aegypti*. Overall, I enjoyed reading it, and I recommend publication in Nature Communications as long as these few minor comments below are addressed:

- 1) A recent paper was published in PNAS which describes the generation of *Ae. aegypti* germline expressed Cas9 strains which significantly increase the germline editing events– Li et al. 2017 (PNAS) – I believe this paper would benefit from a discussion/comparison of the two techniques and efficiencies. Li et al. report up to 90% heritable editing efficiency from a few transgenic Cas9 strains – that is – inject 10 eggs and get 9 different mutants - is ReMOT EEF better than that?
- 2) In the results section (lines 84-118) a *Drosophila* yolk protein peptide P2C (41amnio-acid) was characterized and tested for uptake in 6 different mosquito species by fusing to eGFP. The results are clear, however they left me wondering – why did the authors use a *Drosophila* derived import sequence and not a mosquito derived YP import sequence? How conserved is the 41 amino-acid sequence? I think this section would benefit from an additional bioinformatic based figure highlighting the alignments between all yolk proteins comparing *Drosophila* to mosquitoes (any perhaps a few other insects – aphids?) showing the conservation of the 41 AA sequence.
- 3) In the results section (lines 148-151) these concentrations of EERs, sgRNAs, Cas9 are seem very high – I am wondering if the authors noticed any toxicity? I think toxicity results should be included/mentioned.
- 4) In the results section (Line 183-191) I was initially confused to how these results are different from the previous section (lines 132-133). Then I realized that the previous section was testing for single allele disruption (WT females X White eye males), whereas the latter section is looking for bi-allelic disruption (WT females X WT males). I recommend describing these sections in terms of alleles – would make it a bit clearer for the reader.

Reviewer #2 (Remarks to the Author):

The authors report on a new method to ensure CRISPR-mediated germline transformation in mosquitoes through the use of receptor-mediated uptake of Cas9 RNPs into the developing oocyte. The experimental approach, including using YP1 as a ligand, trimming it to its minimum functional domain through deletion analysis and coupling with a reagent to ensure endosomal release is quite neat. Though the authors claim that the system could be easily adapted to a wide range of arthropods, including some that are particularly refractory to transgenesis, they limit their analysis to showing proof of principle to a mosquito species for which there is already an established protocol for transgenesis through embryo microinjection. I would accept the authors claim that the system is significantly easier to perform and requires less infrastructure than embryo microinjection, the current standard technique. The authors make some nice, well backed-up observations on the fact that both maternal chromosomes in the oocyte and paternal chromosomes in the zygote are susceptible to the activity of Cas9 RNPs injected into the mother. That said though, these experiments were performed by targeting an eye colour gene whose disruption produces an easily scoreable visible phenotype. The authors do not show any evidence for, nor do they discuss at all, the possibility of knockin through homology-directed repair that would allow the easy marking of null alleles through the inclusion of a dominant selectable marker. In the absence of this, relative gains in ease of injection and delivery of nuclease may be offset by the significantly more laborious screening involved.

Rather than list every comment in a detailed list I have included my annotated copy of the text and figures. Some of the more substantial points are also listed below, as well as being on the text:

I found the results section, particularly describing the editing experiments, quite hard to follow. I would suggest moving some sections to supplementary material (eg lines 120-127) and re-naming of section headings ("Gene knockout in wild type mosquitoes by ReMOT Control" on line 182 is also misleading)

Accepting the issue of whether chloroquine as EER seems to improve efficiency, would be a lot easier if it were clearer how many injections sets had been performed without EER. As I understand it the vast majority of injections were performed with chloroquine but germline transformants were also recovered from the non-EER condition. The differential testing depth of the various candidate EERs, or even 'no EER', makes conclusions difficult here. In this section, and throughout, the presentation of the data (e.g Table S3) does not always permit an estimation of the intra-experimental variation – at a minimum simple standard error associated with the mean values would help.

Line168 "Interestingly, for both P2C-Cas9 and P2C-EGFP-Cas9, all mosaics were produced from females injected at later time points post-bloodmeal; females injected at earlier time points produced only white-eye offspring" While this may be a real difference I am unconvinced that you have sufficient numbers to draw this conclusion: 26 'white' and 0 mosaic at 24h versus 30 'white' and 7 mosaic at 54h.

The case for the technology being transferrable to other insects is well made on paper (lines 246-249), though is a little thin in the absence of experimental confirmation. Likewise the speculation on transfer of the technology to mammals (lines 254-258) is just a speculation too far at this stage.

Line 260 "Although GEF was higher for traditional embryo injections compared to ReMOT control, the effort involved (EEF) was substantially higher." The authors make much of these terms GEF and EEF but what about the relative merits of this - it might be for many more laborious to have to

screen and identify low numbers of mutants in a background predominantly wild type. For example, where you don't have a pre-made null allele against which to balance your introduced mutations in the offspring, the efficiency with which you can recover is pretty low. In this case you still had an easily visible phenotype by virtue of the eye colour gene being disrupted - this will not be the case for targeting novel genes of interest, the majority of which will not have easily visible phenotypes. At this point it becomes relevant whether the technology can permit knockins using exogenously provided dominant marker couched in homologous sequence to target site as template for repair.

Figure 3

I found it very difficult to understand if the separate editing events listed all derived from several independent experiments. Panel C doesn't seem to help much and doesn't enlighten on their provenance.

In panel D It is curious that all of the alleles generated at the 519 target site seem to bear the exact same 8bp deletion. This does not seem to be discussed in the text. It is possible that this particular deletion is mechanistically favoured due to microhomology-mediated repair either side of break, coupled with the fact that screening for mutant phenotype should ensure out of frame mutations.

There are also two mistakes in panel D - ko54 has the same 8 bp deletion yet is aligned as if it is different. The PAM sequence underlined at the target site 519 is incorrect.

Reviewer #3 (Remarks to the Author):

Targeted delivery of CRISPR-associated endonuclease 9 (Cas9) into arthropod ovaries for heritable gene editing
Chaverra-Rodriguez, et al

The authors describe a novel method for performing gene editing in arthropods, here using mosquitoes (namely *Ae. aegypti*), by intrathoracically inoculating adult females with a modified Cas9. They have identified a peptide from the *Drosophila melanogaster* yolk protein 1 which can mediate Cas9 into the ovaries of said females and when co-injected with an endosomal release reagent (EER) is able to target DNA, dubbed ReMOT (Receptor-mediated ovary transduction of cargo).

Initial experiments in *An. gambiae* with various DmYP1-EGFP fusions focus in on a 41 amino acid peptide (P2C) which carries EGFP into the ovaries of several mosquito species (Fig2). Further experiments optimized the timing of injection post-blood meal, EER and concentration thereof for *Ae. aegypti*. Optimized conditions and a previously published sgRNA target site within the KMO gene, which yields a recessive white-eyed phenotype were then used for a set of experiments described mainly in Table 1 and Fig3. White-eyed progeny from females injected with P2C-Cas9 and to a lesser extent with P2C-EGFP-Cas9 were obtained by a relatively small (compared to embryonic microinjection) number of injected females. Importantly editing of both the maternal and paternal chromosomes was demonstrated by crossing schemes and confirmed by molecular analysis indicating the complex is active for some time post-injection (more than 24hrs). This concept is clearly translatable to other mosquito species, with *Culex*, *Anopheles*, and *Aedes* genera demonstrated here.

While vitellogenin and yolk-proteins and their receptors have been well studied, this report identifies a novel utilisation of these proteins to shuttle nucleases into the oocytes for the means of genetic manipulation of the germline.

Although the bulk of the experiments were performed in *Ae. aegypti* the authors extrapolate the findings into other species, which is where the true significance to this work applies. Researchers working with organisms which cannot be genetically modified by means of embryo microinjection

could benefit from this technique. Although for those currently using embryo microinjection I see no clear advantage, even though the authors claim a higher effort efficiency or "EEF". This percentage value placed on the effort of injection does not take into account the effort for screening. As shown in Table 2, just comparing the P2C-Cas9 in embryos vs females, the amount of effort injecting 399 embryos vs 142 females is certainly more, I would argue the effort to screen 102 vs 2850 larvae to be significantly less, although unskilled compared to injection.

So, although the percentage of KO is higher per injection it is substantially lower per larvae screened (20.6% for embryos, 1.5% for females). The concentrations of Cas9 protein and sgRNA required for this technique is also ten times higher than that required for standard CRISPR/Cas9 embryo injections and this is barely acknowledged.

Figure 1 provides a clear overview of the procedure and all components involved. The immunofluorescence assay shown in Figure 2E would benefit from a more precise description in the figure legend or on the figure itself. A simple Cas9=Red, DAPI=blue, etc. would suffice. Figure 3 is sufficient and shows representatives of observed phenotypes and sequencing analysis. Note there is an inconsistency between the indicated PAM site for 519 between B and D.

Table 1 presents the bulk of the data of this manuscript and this data is further broken down in Supplemental Tables S3, S4, and S5, same for Tables 2 and S2. I am not convinced of the significance of the hundredth decimal place of a percentage and this already busy table (Table1) could benefit from simplification. Nomenclature referring to the generation is unusual and could be misinterpreted, conventionally the G0 individual is that which is injected rather than its progeny. Evidence indicating editing of maternal and paternal chromosomes is presented, and it appears that conditions differ between the two, with maternal editing more frequent with earlier injection times and paternal editing only presented for a later time of injection (24hr for maternal and 54hr for paternal). Also of note editing of both maternal and paternal alleles was observed, although at a much lower efficiency (1.7%EEF, 0.07%GEF).

Data presented in Table 2 for female injections appears to be the sum of the data presented in Table 1 except for P2CEGFPCas9 (3 white and 8 mosaic in Table 2, 3 mosaic and 8 white in Table 1), for this dataset the %EEF and %GEF do not reflect the values presented here (11KO/305 injected =3.6% and 11/3117larvae = 0.35%). Data presented in Table S3 has been summarized over many trials with various conditions often resulting in a small number of KO progeny, it could be more useful to break-down these trials further by injection conditions to better represent which conditions lead to KO events.

Standard and appropriate methods were carried out for the bulk of the experiments. Methods relating to the endosomal release reagents are non-existent other than what is provided in the legend for Table S4. Figure S2A depicts 5 constructs for expression only 4 of which are shown in the Coomassie, what happened to P2C-Cas9? More generally, for a methods paper such as this, it is crucial that the reader can see all details and reconstruct what was done from the text. I do not think that this is possible with the present ms.

Data support the conclusion that this technique can be used to shuttle CRISPR/Cas9 into the oocytes of mosquitoes and likely other dipterans. Support for the conceptual extension of this technique into many other species including mammals is minimal with only evidence of P2C-EGFP in the ovaries of 6 mosquito species. It is assumed that P2C could also carry a functional CRISPR/Cas9 complex, although no data is presented for any species other than *Ae. aegypti*. While embryo microinjection is a specialist technique and can also be a financial barrier to genetic manipulation, it can be overcome. The highest impact lies in the capability for genetic manipulation in organisms in which it is currently difficult or impossible.

Because all experiments were performed in mosquitoes and predominantly *Ae. aegypti*, it is difficult to determine how far reaching this technique could be. Experiments in an organism which has not previously been genetically modified would go a long way in supporting this conclusion.

Response to reviewers comments on “Targeted delivery of CRISPR-associated endonuclease 9 (Cas9) into arthropod ovaries for heritable germline gene editing”

Duverney Chaverra-Rodriguez, Vanessa M. Macias, Grant L. Hughes, Sujit Pujhari, Yasutsugu Suzuki, David R. Peterson, Donghun Kim, Sage McKeand, Jason L. Rasgon

We thank the reviewers for their thoughtful review of our manuscript. We have summarized the reviewers comments and criticisms below (in bold) and discussed the revisions made to the original manuscript in response to these criticisms (in plain text).

In addition to the revisions made in response to reviewers, We have made a few additional extra revisions improve the presentation of data. These include the following:

1) Taking into consideration a combination of comments from the reviewers, figure 1 has been completely redesigned and includes all information displayed on the original figure, but now in addition to the mechanism of ReMOT Control we have integrated into the illustration a comparison of ReMOT Control and embryo microinjection approaches to targeted mutation including reagents, equipment, time required and screening limitations (which were all categories requested to be emphasized by reviewers), and a model explaining our hypothesis regarding mosaicism and timing of injection.

2) Figure layouts have been modified for clarity and visual quality.

3) Table 2 has been removed to avoid miscomparison of different injection experiments (both adult and embryo and both biallelic and monoallelic). Supplementary table 2 includes the embryo injection data used to generate the KMO460 knockout line used for outcrossing.

Reviewer 1. Omar Akbari

Lack of discussion of results with regard to recent advancements made by Li *et al.* in the same species.

Recently, Cas9-expressing lines are described by Li *et al.* in PNAS that increase the rate of HDR to 90% using embryo injection. The rate of mutation using ReMOT Control reported here is not this high, as reflected in the reported EEF, but the primary advantage of ReMOT Control is the non-technical method used to generate mutants. Ten embryo injections still requires microinjection equipment and training, where a mutation can be generated by ReMOT Control without these requirements, so this work opens up gene-editing capability in labs that aren't currently equipped to use microinjection, and paves the way for utilizing gene-editing in species that are not amenable to microinjection. It is likely that a combination of Cas9 expression and methods to deliver editing moieties to the germline will result in a more robust execution of ReMOT Control and this work is underway. A discussion of these points have been added to the text

In the results section (lines 84-118) a *Drosophila* yolk protein peptide P2C (41 amino-acid) was characterized and tested for uptake in 6 different mosquito species by fusing to eGFP. The results are clear, however they left me wondering – why did the authors use a *Drosophila* derived import sequence and not a mosquito derived YP import sequence?

The *Drosophila* yolk protein peptide was previously demonstrated to be efficiently taken up into mosquito ovaries (Bownes *et al.*, 2002) and is small relative to the vitellogenins in mosquitoes, so it was the first in we tested to mediate the uptake of EGFP. This was efficient, so we

proceeded to use it to develop ReMOT Control in *Ae. aegypti*. We have been exploring derivatives of Vg, but these experiments are not nearly as advanced as our work with P2C and will be explored in future publications.

How conserved is the 41 amino-acid sequence? I think this section would benefit from an additional bioinformatic based figure highlighting the alignments between all yolk proteins comparing *Drosophila* to mosquitoes (any perhaps a few other insects – aphids?) showing the conservation of the 41 AA sequence.

The 41- amino acid fragment (P2C) is not homologous to any known vitellogenin sequences either in arthropods or vertebrates. However, it shows homology with several protein sequences from lipases in vertebrates and yolk proteins in Cycloraphan dipterans. A discussion of this has been added to the text.

In the results section (lines 148-151) these concentrations of EERs, sgRNAs, Cas9 are seem very high – I am wondering if the authors noticed any toxicity? I think toxicity results should be included/mentioned.

We collected data on survival and oviposition following injection, however we did not perform analogous control injections and so cannot perform a formal analysis. General survival between 60-80% provided sufficient offspring to consistently attain mutants. We have added text to include a discussion of survival.

In the results section (Line 183-191) I was initially confused to how these results are different from the previous section (lines 132-133). Then I realized that the previous section was testing for single allele disruption (WT females X White eye males), whereas the latter section is looking for bi-allelic disruption (WT females X WT males). I recommend describing these sections in terms of alleles – would make it a bit clearer for the reader.

The language has been changed in these sections to increase clarity.

Reviewer #2:

Textual comments:

All text corrections made. Additionally, comments on the text that required responses have been copied here and responses are provided.

Justification for using this species seems weird, when you used *An. gambiae* for initial yp1 validation

An. gambiae was used initially because the in an old paper *Drosophila* yolk protein had been shown to enter the ovaries of *An. gambiae*. After validating that these results were true, and determining the precise region of the protein responsible for uptake, we switched further work to *Ae. aegypti*, since methods for transformation by Cas9 are more advanced in this species (which is also easier to rear and manipulate in the lab), allowing us to examine guide RNA/target site pairs that had already been validated. These motivations are now described in the text.

Regarding EEF and GEF: “this seems out of place here- probably more of an editing comment but I think it makes much more sense to show us conclusively that it works, and then later make your arguments about efficiencies relative to other approach”

In this paragraph we begin reporting our results, including both numbers of mutants and the related frequency values (EEF and GEF), so describing the ways that we are portraying frequency is necessary to fully understand the values in the table.

In the section title “Heritability of *kmo* mutants,” the reviewer states: “these are presumably from the 9/11161 (i.e wt x wt), which themselves were obtained as progeny from non-white G₀ (rather than progeny from G_{minus}1)? If so these are almost bound to show heritability as they show the trait two generations removed from the original injected female. It still may be the case, in the G₀ offspring of wt x *kmo*460, that you have apparently white individuals that do not show full heritability of the trait

The G₀ mutants obtained ***are not*** from the WTxWT intercross, as the Reviewer presumes, but from outcrossed G₋₁ injected females as stated in the text.

Main comments:

Both Reviewers 1 and 2 remark that a discussion of the potential use of HDR to improve screening of a mutation with no visible phenotype following ReMOT Control is missing.

The use of *kmo* in our proof of concept experiments allowed us to develop ReMOT Control, but in further use of this technology, it is likely that researchers will desire mutations in genes with no visible phenotype. In such experiments, mutants will have to be detected using molecular methods, such as T7 assays or sequencing of amplicons from the target gene. These methods are already in place for microinjection experiments (Basu et al 2015, Kistler et al 2015, Dong et al. 2015, DeGennaro et al 2013, McMeniman *et al.* 2014) but are laborious. ReMOT Control would certainly benefit from HDR-mediated integration of a visual marker at the target site, which would require the delivery of donor DNA into the ovary. Delivery of DNA to the mosquito germline by adult injection has been attempted by several groups since it would increase the ease of all types of genetic manipulation, including transposon-mediated and integrase-mediated transgenesis, but no method that supports transgenesis has been reported. We are currently expending considerable resources to develop methods to do this. However, detection of mutations without this advancement is done regularly by other groups using molecular methods. A discussion of this and the impact of laborious molecular screening have been added to the discussion section of the revised manuscript.

I found the results section, particularly describing the editing experiments, quite hard to follow. I would suggest moving some sections to supplementary material (eg lines 120-127) and re-naming of section headings (“Gene knockout in wild type mosquitoes by ReMOT Control” on line 182 is also misleading)

The text has been significantly modified to improve clarity, including revisions to the sections headings, as suggested by the reviewer. We feel that the information presented on the establishment of the white-eye line is essential to the information presented in figure 4 as the knockout line was generated with different guide RNA (at site 460) and so allows us to distinguish mutations induced by ReMOT Control from those of the parental white-eye line. This

has been clarified in the revised manuscript

Accepting the issue of whether chloroquine as EER seems to improve efficiency, would be a lot easier if it were clearer how many injections sets had been performed without EER. As I understand it the vast majority of injections were performed with chloroquine but germline transformants were also recovered from the non-EER condition. The differential testing depth of the various candidate EERs, or even ‘no EER’, makes conclusions difficult here. In this section, and throughout, the presentation of the data (e.g Table S3) does not always permit an estimation of the intra-experimental variation – at a minimum simple standard error associated with the mean values would help.

Table S4 has been edited and shows the number of mosquito injected with no EER (as before) and the number of trials (usually 20-30 mosquitoes that yielded 1 mutant each time with chloroquine). All of the maternal editing “no EER” experiments resulted in zero mutants. A single mutant using no EER was recovered for paternal injection experiment, so in reality there isn’t much experimental variation at all.

Line168 “Interestingly, for both P2C-Cas9 and P2C-EGFP-Cas9, all mosaics were produced from females injected at later time points post-bloodmeal; females injected at earlier time points produced only white-eye offspring” While this may be a real difference I am unconvinced that you have sufficient numbers to draw this conclusion: 26 ‘white’ and 0 mosaic at 24h versus 30 ‘white’ and 7 mosaic at 54h.

This remark referenced by the reviewer on line 168 is not a conclusion, but an observation. We speculate that this may reflect a biological difference the way that RNPs are internalized and access the genomic DNA at different stages of oocyte and embryo development, but we do not draw conclusions based on these data. The statement has been elaborated on to clarify this.

The case for the technology being transferrable to other insects is well made on paper (lines 246-249), though is a little thin in the absence of experimental confirmation. Likewise the speculation on transfer of the technology to mammals (lines 254-258) is just a speculation too far at this stage.

This comment is similar to a comment also made by Reviewer 2; we will answer both here. We agree that we do not have data to support a conclusion that this can be transferred to mammals, so we were intentional to only speculate that specific, though not conserved, ovary uptake mechanisms in both insects and mammals could be analogously used for genetic manipulation. Such far-reaching speculation is the seed of widespread innovation and we believe that this part of the discussion will enhance efforts to improve genetic engineering technology in a diversity of organisms.

Figure 3

I found it very difficult to understand if the separate editing events listed all derived from several independent experiments. Panel C doesn’t seem to help much and doesn’t enlighten on their provenance.

Figure 3 summarizes the visual phenotypes (panels A and D) and sequence genotypes (panel C and F) seen in knock-down experiments and is intended to be representative (as noted by reviewer 3), not an exhaustive detailing of each knock-out event, but to present examples of all phenotypes and genotypes detected. The figure legend has been modified to clarify this. We agree that panel C is superfluous and it has been removed.

In panel D It is curious that all of the alleles generated at the 519 target site seem to bear the exact same 8bp deletion. This does not seem to be discussed in the text. It is possible that this particular deletion is mechanistically favoured due to microhomology-mediated repair either side of break, coupled with the fact that screening for mutant phenotype should ensure out of frame mutations.

We noticed this too and were remiss to include a discussion of this observation. Such a discussion has been added.

There are also two mistakes in panel D – ko54 has the same 8 bp deletion yet is aligned as if it is different. The PAM sequence underlined at the target site 519 is incorrect.

Corrected.

Reviewer #3 (Remarks to the Author):

Researchers working with organisms which cannot be genetically modified by means of embryo microinjection could benefit from this technique. Although for those currently using embryo microinjection I see no clear advantage, even though the authors claim a higher effort efficiency or “EEF”. This percentage value placed on the effort of injection does not take into account the effort for screening. As shown in Table 2, just comparing the P2C-Cas9 in embryos vs females, the amount of effort injecting 399 embryos vs 142 females is certainly more, I would argue the effort to screen 102 vs 2850 larvae to be significantly less, although unskilled compared to injection. So, although the percentage of KO is higher per injection it is substantially lower per larvae screened (20.6% for embryos, 1.5% for females).

The EEF value is not intended to consider and compare the overall effort of the technology, but instead to provide an editing efficiency value that can be compared between adult and embryo injections, since different generations are in these two approaches. The screening can be directly compared between the two methods, as this reviewer demonstrates in his/her comment so we saw no need to devise a new value to represent screening. However, we have included a more explicit emphasis on the increased effort of screening required using ReMOT Control both in the text and in the revised Figure 1.

The concentrations of Cas9 protein and sgRNA required for this technique is also ten times higher than that required for standard CRISPR/Cas9 embryo injections and this is barely acknowledged.

This is acknowledged in the discussion section; however the higher concentrations injected into females are offset by the number of G₀ oocytes targeted in *one* adult injection (more than a hundred, but on average 20 surviving G₀ offspring/female) compared to one G₀ embryo targeted per microinjection. Even more, the volume of liquid material required for embryo microinjections

is substantially more than what is actually delivered to the embryos, since needles have to be changed regularly during microinjection; most of the volume needed is not actually injected. In order to provide a more thorough treatment of this subject, additional text has been added to the discussion and Figure 1 has been revised to include the comparison.

The immunofluorescence assay shown in Figure 2E would benefit from a more precise description in the figure legend or on the figure itself. A simple Cas9=Red, DAPI=blue, etc. would suffice.

Information added to figure legend, as suggested.

Figure 3 is sufficient and shows representatives of observed phenotypes and sequencing analysis. Note there is an inconsistency between the indicated PAM site for 519 between B and D.

Corrected.

Table 1 presents the bulk of the data of this manuscript and this data is further broken down in Supplemental Tables S3, S4, and S5, same for Tables 2 and S2. I am not convinced of the significance of the hundredth decimal place of a percentage and this already busy table (Table1) could benefit from simplification.

This table has been simplified including fewer significant digits.

Nomenclature referring to the generation is unusual and could be misinterpreted, conventionally the G₀ individual is that which is injected rather than its progeny.

The nomenclature developed for transgenesis by embryo microinjection did indeed indicate G₀ as the injected individual (injected as an embryo), however we disagree that our naming is unconventional since we have assigned generation number to align biologically with what is traditionally used in microinjection. Using ReMOT Control, gene manipulation moieties are also delivered to the G₀, albeit indirectly through the maternal hemolymph. We have included an illustration of this in our revamping of Figure 1 that clarified this.

Data presented in Table 2 for female injections appears to be the sum of the data presented in Table 1 except for P2CEGFPCas9 (3 white and 8 mosaic in Table 2, 3 mosaic and 8 white in Table 1), for this dataset the %EEF and %GEF do not reflect the values presented here (11KO/305 injected =3.6% and 11/3117larvae = 0.35%).

Corrected.

Data presented in Table S3 has been summarized over many trials with various conditions often resulting in a small number of KO progeny, it could be more useful to break-down these trials further by injection conditions to better represent which conditions lead to KO events.

Such a breakdown is represented in Tables S3 and S4. Additionally at this stage the impact of

EERs on ReMOT Control was unknown; these experiments were exploratory and the LD50 only preliminary estimations in order to aid in the search for an effective EER. Once C]chloroquine was identified as an effective EER, further optimization proceeded with this chemical. The text has been edited to clarify this.

Figure S2A depicts 5 constructs for expression only 4 of which are shown in the Coomassie, what happened to P2C-Cas9?

The gel picture in Figure S2A has been replaced with one that includes P2C-Cas9 in addition to the other proteins.

More generally, for a methods paper such as this, it is crucial that the reader can see all details and reconstruct what was done from the text. I do not think that this is possible with the present ms.

We agree with Reviewer 3 that a thorough treatment of the methods should allow for replication by other groups. While a large set of methods are presented here, including DNA and protein engineering and mosquito handling, these methods are straight forward and widely used molecular biology and mosquito manipulation techniques. All methods, recipes and materials used are presented in detail in the methods sections including recipes, reagents sources and mosquito husbandry and injection conditions. As the reviewer did not specifically identify what is missing, we are unable to address this comment, but will be happy to address a specific concern.

REVIEWERS' COMMENTS:

Reviewer #1 (Remarks to the Author):

The authors adequately addressed all of my concerns.

Omar S. Akbari
Assistant Professor
Cell and Developmental Biology Section
Division of Biological Sciences
University of California, San Diego
9500 Gilman Drive
La Jolla, CA 92093-0349
www.akbarilab.com

Email: oakbari@ucsd.edu
Phone: (858) 246-0640

NOTICE: This email message is for the sole use of the intended recipient(s) and may contain confidential and privileged information. Any unauthorized review, use, disclosure or distribution is prohibited. If you are not the intended recipient, please contact the sender by reply email and destroy all copies of the original message.

Reviewer #2 (Remarks to the Author):

I think the manuscript is markedly improved. I am glad the discussion of target site choice and restricted set of null alleles is included and there is a fairer and more extended discussion of the relative merits of the technology, and some of its limitations as it currently stands in terms of HDR and labour-intensive screening etc.

I also still think the description of the crossing procedures to tease out the mutations, and what these show, could be clearer, but maybe it's just me (a figure for the cross and genotypes would certainly help). However, I don't want to hold the manuscript up any further. I just think that some of the results are cool and it would be better to make them as interpretable to as wide an audience as possible. For example Section at 186 Bi-allelic knockout by ReMOT Control – I believe this section has been re-named (perhaps at my suggestion) but the data here don't really confirm bi-allelic mutations do they? At least as I understand the term, bi-allelic, when referring to nuclease-induced mutations, means that both alleles in a diploid cell are mutated by the nuclease. What is reported here cannot distinguish separate monoallelic mutations that were brought together by simple crossing. The only suggestion of biallelic mutation caused by the nuclease here would be the single mosaic individual, which died, and may have represented some somatic biallelic activity. What is reported here cannot distinguish separate monoallelic mutations that were brought together by simple crossing. The only suggestion of biallelic mutation caused by the nuclease here would be the single mosaic individual, which died, and may have represented somatic biallelic activity

I also still think that the speculation on the application to vertebrates is too long but this is an editorial thing so I am not going to insist.

Couple of small points

Line 167 use 'gene editing of the paternal allele' rather than paternal gene editing

Syntax on lines 126-128 needs sorting

Reviewer #3 (Remarks to the Author):

"Targeted delivery of CRISPR-associated endonuclease 9 (Cas9) into arthropod ovaries for heritable germline gene editing"

Duverney Chaverra-Rodriguez, Vanessa M. Macias, Grant L. Hughes, Sujit Pujhari, Yasutsugu Suzuki, David R. Peterson, Donghun Kim, Sage McKeand, Jason L. Rasgon

My major hesitation which was not addressed in the rebuttal relates to the extrapolation of the data presented in *Ae. aegypti*, into other species up to mammals as stated previously.

From the review of the initial submission: "Although the bulk of the experiments were performed in *Ae. aegypti* the authors extrapolate the findings into other species, which is where the true significance to this work applies. Researchers working with organisms which cannot be genetically modified by means of embryo microinjection could benefit from this technique...

Support for the conceptual extension of this technique into many other species including mammals is minimal with only evidence of P2C-EGFP in the ovaries of 6 mosquito species. It is assumed that P2C could also carry a functional CRISPR/Cas9 complex, although no data is presented for any species other than *Ae. aegypti*. While embryo microinjection is a specialist technique and can also be a financial barrier to genetic manipulation, it can be overcome. The highest impact lies in the capability for genetic manipulation in organisms in which it is currently difficult or impossible. Because all experiments were performed in mosquitoes and predominantly *Ae. aegypti*, it is difficult to determine how far reaching this technique could be. Experiments in an organism which has not previously been genetically modified would go a long way in supporting this conclusion." In lines 281-290 it is suggested that ReMOT Control to work in other species the ligand/receptor, time of injection, endosomal release reagent, and amount of RNP would need to be optimized indicating the concept of this technique could be translated to other species although the specifics may not, which limits the impact considerably.

Minor issues:

Line 137 mention of supplementary table 3 should be 4

Line 145 mention of supplementary table 4 should be 3

Line 257 "recommended 66% recommendation"

Line 333 a suggestion for pulling the needle would be helpful as the comparison to embryonic microinjection states a laser puller is not required

Line 662 Two versions of the Supplemental figure titles and legends has been provided, neither of which seems to match the tables/figures

Line 666 I believe this should be related to figure 4

Line 673 title does not match the table

Table S4 A more informative way to present the % for those experiments which did not result in a KO event would be as less than __%, in other words the efficiency would have to be lower than a certain percentage to be detectable by this experiment.

Referees response

Please see our responses to the reviewers' second round of comments in bold below.

Reviewer #1 (Remarks to the Author):

The authors adequately addressed all of my concerns.

Reviewer #2 (Remarks to the Author):

I think the manuscript is markedly improved. I am glad the discussion of target site choice and restricted set of null alleles is included and there is a fairer and more extended discussion of the relative merits of the technology, and some of its limitations as it currently stands in terms of HDR and labour-intensive screening etc.

I also still think the description of the crossing procedures to tease out the mutations, and what these show, could be clearer, but maybe it's just me (a figure for the cross and genotypes would certainly help). However, I don't want to hold the manuscript up any further. I just think that some of the results are cool and it would be better to make them as interpretable to as wide an audience as possible. For example Section at 186 Bi-allelic knockout by ReMOT Control – I believe this section has been re-named (perhaps at my suggestion) but the data here don't really confirm bi-allelic mutations do they? At least as I understand the term, bi-allelic, when referring to nuclease-induced mutations, means that both alleles in a diploid cell are mutated by the nuclease. What is reported here cannot distinguish separate monoallelic mutations that were brought together by simple crossing. The only suggestion of biallelic mutation caused by the nuclease here would be the single mosaic individual, which died, and may have represented some somatic biallelic activity.

We agree, thank you, and have included a new figure, Figure 4, that we believe addresses this.

I also still think that the speculation on the application to vertebrates is too long but this is an editorial thing so I am not going to insist.

This discussion removed.

Couple of small points

Line 167 use 'gene editing of the paternal allele' rather than paternal gene editing

We changed the sentence as suggested

Syntax on lines 126-128 needs sorting

We modified the whole paragraph to explain our outcrossing strategy to detect gene editing activity

Reviewer #3 (Remarks to the Author):

“Targeted delivery of CRISPR-associated endonuclease 9 (Cas9) into arthropod ovaries for heritable germline gene editing”

Duverney Chaverra-Rodriguez, Vanessa M. Macias, Grant L. Hughes, Sujit Pujhari, Yasutsugu Suzuki, David R. Peterson, Donghun Kim, Sage McKeand, Jason L. Rasgon

My major hesitation which was not addressed in the rebuttal relates to the extrapolation of the data presented in *Ae. aegypti*, into other species up to mammals as stated previously.

From the review of the initial submission: “Although the bulk of the experiments were performed in *Ae. aegypti* the authors extrapolate the findings into other species, which is where the true significance to this work applies. Researchers working with organisms which cannot be genetically modified by means of embryo microinjection could benefit from this technique...

Support for the conceptual extension of this technique into many other species including mammals is minimal with only evidence of P2C-EGFP in the ovaries of 6 mosquito species. It is assumed that P2C could also carry a functional CRISPR/Cas9 complex, although no data is presented for any species other than *Ae. aegypti*. While embryo microinjection is a specialist technique and can also be a financial barrier to genetic manipulation, it can be overcome. The highest impact lies in the capability for genetic manipulation in organisms in which it is currently difficult or impossible.

Because all experiments were performed in mosquitoes and predominantly *Ae. aegypti*, it is difficult to determine how far reaching this technique could be. Experiments in an organism which has not previously been genetically modified would go a long way in supporting this conclusion.”

In lines 281-290 it is suggested that ReMOT Control to work in other species the ligand/receptor, time of injection, endosomal release reagent, and amount of RNP would need to be optimized indicating the concept of this technique could be translated to other species although the specifics may not, which limits the impact considerably.

We did address this comment along with a similar comment to another reviewer: from our previous response, though incompletely: *“This comment is similar to a comment also made by Reviewer 2; we will answer both here. We agree that we do not have data to support a conclusion that this can be transferred to mammals, so we were intentional to only speculate that specific, though not conserved, ovary uptake mechanisms in both insects and mammals could be analogously used for genetic manipulation. Such far-reaching speculation is the seed of widespread innovation and we believe that this part of the discussion will enhance efforts to improve genetic engineering technology in a diversity of organisms.”*

We lumped these two reviewers’ criticisms together, but should not have, as Reviewer 2 believed the extensions to insects reasonable, but, as stated above, Reviewer 3 is more skeptical of this. We have minimized the text suggesting this technology be extended to oviparous vertebrates and mammals. With regard to

other mosquito species and possibly other insects: a major innovation in this paper, the delivery of material to the oocyte, we do demonstrate that this delivery works in several other mosquito species without changing the quality of the P2C fragment at all. It is reasonable to suspect that we can find similar effectors in other insect species and along with our proof of principle that this delivery can be paired with editing, we believe this sufficient motivation to explore using ReMOT Control in taxa where gene-editing is not as straight forward or not possible.

Minor issues:

Line 137 mention of supplementary table 3 should be 4

Thank you, we changed it

Line 145 mention of supplementary table 4 should be 3

Thank you, we changed it

Line 257 “recommended 66% recommendation”

Deleted “recommendation” it is now on line 271

Line 333 a suggestion for pulling the needle would be helpful as the comparison to embryonic microinjection states a laser puller is not required

On line 326 we included: “a standard (non-laser) capillary puller or a flame can be used to pull glass needles or affordable pre-pulled glass needles are available from commercial vendors (in the absence of a needle puller)”

The following have been corrected:

Line 662 Two versions of the Supplemental figure titles and legends has been provided, neither of which seems to match the tables/figures

Line 666 I believe this should be related to figure 4

Line 673 title does not match the table

Table S4 A more informative way to present the % for those experiments which did not result in a KO event would be as less than __%, in other words the efficiency would have to be lower than a certain percentage to be detectable by this experiment.

If we understand the comment correctly, the reviewer is suggesting that 0.00% is not an appropriate way to represent an efficiency that may not be absolutely zero, but may be very small (e.g. 0.0001) and so undetectable in our test. However, since the number tested is reported in the table and two significant digits are present after the decimal point we are implying that the efficiency is zero within our tested number.

** See Nature Research's author and referees' website at www.nature.com/authors for

information about policies, services and author benefits